# Digital Rock Mechanical Properties by Simulation of True Triaxial Test: Impact of Microscale Factors

**Wenjie Ma [1], Yongfei Yang [1],\*, Wendong Yang [2], Changran Lv [1], Jiangshan Yang [1], Wenhui Song [1], Hai Sun [1], Lei Zhang [1], Kai Zhang [1] and Jun Yao [1]**

1   School of Petroleum Engineering, China University of Petroleum, Qingdao 266580, China
2   College of Pipeline and Civil Engineering, China University of Petroleum, Qingdao 266580, China
\*   Correspondence: yangyongfei@upc.edu.cn

**Abstract:** Complex fractures and pore structures in the rock strongly influence the mechanical properties, and the process from compression to failure is complicated. Under the action of rock stress, pore structure deformation and fractures close or propagate, easily leading to deterioration in the rock mechanical properties until rock failure. Thus, the effects of microscale factors are critical in mechanical properties such as rock strength, elastic modulus, and stress–strain state under the triaxial stress state. It is difficult for physical and mechanical experiments to obtain the qualitative rules of regular structures, but numerical simulation can make up for this defect. In this work, the accuracy of the model was proven through a comparison with previous experimental results. The true triaxial numerical simulation experiments were conducted on representative rocks and natural pore structures. These simulated results revealed that the pore and throat parameters will change abruptly when the particle model volumetric strain is between 0.0108 and 0.0157. When the fracture angle is between 45° and 75°, the fracture has a great influence on the peak stress. The angle between the natural fracture and the fracturing direction should be less than 45° as much as possible. Clay affects the rock strength by influencing the force chains formed by the rock skeleton. Fracturing is easier when the structural clay content is higher than 25%. It is easier to fracture in a direction parallel to the laminated clay when the clay content is below 27%. This work indicates the effects of rock particles, fractures, and clay on the mechanical parameters, providing key fundamental data for further quantifying the fracturing patterns.

**Keywords:** digital rock; mechanical properties; stress–strain state; microscale structures; numerical simulation

## 1. Introduction

Deep underground rocks store a large number of oil and gas resources, and their physical properties are diverse and heterogeneous [1,2]. There are complex fractures and pore structures in the rocks, and the process from compression to the failure of rocks is complicated [3–5]. Under the action of rock stress, pore structures and rock skeleton deformation, fractures close or propagate and connect with other fractures [6,7]. Thus, the effects of microscale factors are critical not only in the mechanical properties, but also in engineering operations such as fracturing and logging [8].

The rock physics experiment is the most basic rock physics research method, and the conventional triaxial test and true triaxial test are simple to operate [9]. According to the engineering design of construction and the environmental conditions, the true triaxial test can change the stresses of three axes to simulate the real stress environment accurately. However, the traditional rock physics experiment has the disadvantages of long experiment period, high cost, poor repeatability, and large error under low porosity and permeability [10].

X−ray CT scanning technology can nondestructively detect the structure inside the object [11]. It provides a very effective method for visualizing a complex three−dimensional

crack's geometry and distribution in rock [12,13]. Through the reconstruction of rock images, the digital rock model obtained can be used for rock simulation research [14]. The numerical methods commonly used in rock mechanics can be divided into the continuous medium method, discrete medium method, and the mixed method of continuous and discrete medium [15–20]. When the simulated area is large enough, even if there are rock joint and fractures, the mechanical behavior of the rock can be reflected by using the continuity method. However, when the simulated area is relatively small, the material discontinuities cannot be ignored, and the existence of rock particles and fractures needs to be considered. Compared with physical tests, numerical simulation can quantitatively and repeatably investigate the influences of various factors on the mechanical properties of the rock.

The mechanical properties of rocks refer to the elasticity, plasticity, elastoplasticity, rheology, brittleness, toughness, heating, and other mechanical properties of rocks under stress [21]. External factors such as temperature, humidity, loading–unloading conditions, and rock confining pressure affect the mechanical properties [22–24]. The properties vary greatly due to the age at which the particles, fractures, and clays were formed [25].

Rock is formed by the combination of mineral particles with different particle sizes. Thus, the physical and mechanical properties of the rock are bound to be affected by mineral particles, and the particle size of mineral particles is one of the non−negligible factors [26]. The strength, physical, and mechanical properties, characteristics of the acoustic emission and cooling rate after the high temperature of rocks with different particle sizes are significantly different [27–30]. However, there have been few quantitative studies on the different particle sizes.

Rock fractures close or propagate and coalescence, and pore structures deform to resist stress. The effect of fractures on the mechanical properties makes them a major consideration during fracturing. The influences of prefabricated crack morphology, fatigue crack growth, and naturally cemented fractures were studied [31–33]. There have been many studies on the number of prefabricated fractures, but the angle and aperture of fractures should not be ignored.

The presence of clay in rock, which is different from the rock skeleton, will make the "force chain" of the particle model incomplete, and the mechanical properties of the rock deteriorate [34]. Compaction, types of clay, content of clay, distribution of clay, and creep characteristics make the influence of clay on the mechanical properties vary [35–38]. However, there are few quantitative simulations of clay in digital rocks. The effects of silt and clay on the mechanical properties are similar, and a large number of studies have focused on triaxial compression tests of gas hydrate−bearing sediments, frozen silt, and coast silt [39–41].

In this work, based on the digital rock, true triaxial test numerical simulation experiments were conducted on representative rocks and natural pore structure. The rock mechanical properties were studied on the microscale to simulate the force state closer to the real situation and explore the change rule. This work indicates the effects of rock particles, fractures, and clay on the mechanical parameters, providing key fundamental data for further quantifying fracturing patterns and guiding hydraulic fracturing to promote oil and gas well stimulation.

## 2. Materials and Methods

### 2.1. Mathematical Model

The finite element method is a basic and mature numerical simulation method for the calculation of the mechanical parameters of digital rock without grid discretization [42]. First, numerical simulation can obtain that the steady state should satisfy the mechanical equilibrium, and the equilibrium equation is as follows:

$$\rho \frac{\partial^2 \mu}{\partial t^2} = \nabla \cdot \sigma + \mathbf{F_v} \tag{1}$$

where $\rho$ is the density in the actual deformed state; $\boldsymbol{\mu}$ is displacement vector; $\sigma$ is stress; and $\mathbf{F_v}$ is the volume force vector. For the linear elastic material, Hooke's law relates the stress tensor to the elastic strain tensor [43]:

$$\sigma = \sigma_i + \mathbf{C} : \varepsilon_{e1} \tag{2}$$

where $\sigma_i$ is the initial stress; $\mathbf{C}$ is the fourth order elastic tensor; and $\varepsilon_{e1} = \varepsilon - \varepsilon_{ine1}$ is the elastic strain, which is computed by a straightforward subtraction of the inelastic strain.

The elastic part should adopt the linear elastic model. For isotropic materials, the fourth order elasticity tensor $\mathbf{C} = \mathbf{C}(E, \nu)$ has only two independent components: Poisson's ratio $\nu$ and Young's modulus $E$. Different elastic moduli can be used, as long as two moduli are defined. The relationship can be expressed as:

$$\nu = \frac{3K - 2G}{2(3K + G)}, E = \frac{9KG}{3K + G} \tag{3}$$

where $K$ is the volume modulus and $G$ is the shear modulus.

The Lagrange strain tensor can be expressed as [44]:

$$\varepsilon = \frac{1}{2}\left[(\nabla\mathbf{u})^{\mathrm{T}} + \nabla\mathbf{u}\right] \tag{4}$$

When small plastic strain is selected as the plasticity model for the plasticity node, the direction of the plastic strain increment is defined by [45]

$$\dot{\varepsilon}_{\mathrm{pl}} = \lambda\frac{\partial Q_{\mathrm{p}}}{\partial\sigma} \tag{5}$$

where $Q_{\mathrm{p}}$ is the plastic potential function; $\lambda$ is a plastic multiplier that depends on the current state of stress and the load history; $\sigma$ is the current state of stress. It shows the relationship between the increment of the plastic strain tensor and the current state of stress.

The plastic multiplier $\lambda$ is determined by the complementarity or Kuhn–Tucker conditions

$$\lambda \geq 0, F_y \leq 0 \text{ and } \lambda F_y = 0 \tag{6}$$

where $F_y$ is the yield function. The yield surface encloses the elastic region defined by $F_y < 0$ and plastic flow occurs when $F_y = 0$.

The rock yield criterion is the Drucker–Prager criterion [46,47]:

$$F_y = \sqrt{J_2} + \alpha I_1 - k \tag{7}$$

where $I_1 = \sigma_1 + \sigma_2 + \sigma_3$ is the first invariant of the stress tensor; $J_2 = \frac{1}{6}\left[(\sigma_1 - \sigma_2)^2 + (\sigma_2 - \sigma_3)^2 + (\sigma_3 - \sigma_1)^2\right]$ is the second invariant of the deviatoric stress tensor; $\alpha = \frac{2\sin\phi}{\sqrt{3}(3-\sin\phi)}$ and $k = \frac{2\sqrt{3}c\cos\phi}{3-\cos\phi}$ are the experimental constants related to the internal friction angle $c$ and cohesion $\phi$ of rocks when matching the criterion in the generalized plane−strain [48].

## 2.2. Physical Model

The true triaxial test method is used to simulate various digital rocks. According to the mechanical equilibrium, any rock in space is subjected to forces in the $x$, $y$, and $z$ directions. Unlike the conventional triaxial test, the true triaxial test is conducted for three principal stresses with different sizes. Figure 1 shows the force diagram.

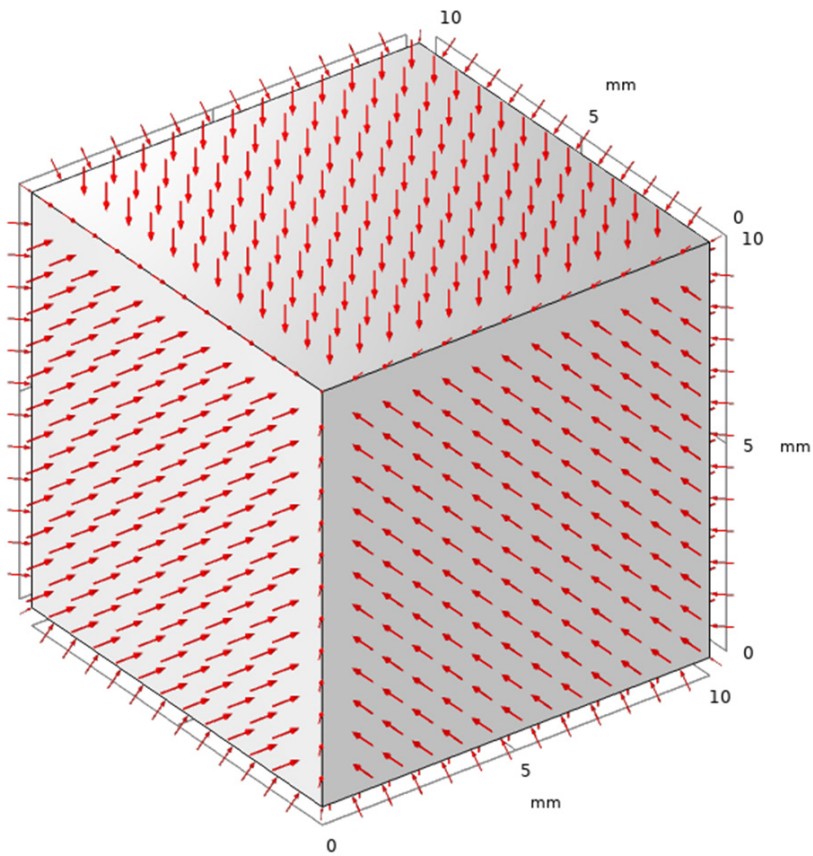

**Figure 1.** True triaxial stress test force diagram.

The triaxial simulation experiments described in this paper were performed using the finite element method. Based on solid mechanics theory, the stress–strain curve, rock stress field distribution, deformation characteristics, and mechanical parameter changes can be obtained. This work applied displacement to the model in three axial directions and calculated the stress. Using the repeatability of the simulation, the stress−strain states were calculated for all triaxial displacement cases of the same model. We then found a case where the axial stress was the greatest in all cases and defined it as the maximum principal stress. In this case, the remaining two axial stresses can be divided into intermediate principal stress and minimum principal stress depending on the value. Change the microscale factors of representative and natural rocks, conduct the true triaxial test simulation, study the qualitative law, and finally obtain the microscale influence law of the digital rock mechanical properties.

### 3. Results and Discussion

*3.1. Model Validation*

In order to verify the accuracy of the simulation of the digital rock mechanical properties, this study used a volcanic rock as an example to conduct the model trial calculation. The volcanic rock has strong heterogeneity, and has micro−fractures and pores. Volcanic rocks are reproduced using CT scanning combined with digital rock technology to reproduce the complex pore space. Based on the constructed digital rock, the accuracy of the simulation was studied.

The device used for CT scanning in this study was the MicroXCT−400. In the process of CT scanning, the sample is rotated to obtain the projection image of different angles, and then the gray image of the internal pore structure of the rock can be obtained. In order to ensure the quality and accuracy of subsequent image processing, the scanning data need to be further processed such as adjusting the display information, determining the minimum representative elementary volume (REV), image denoising, image segmentation, etc.

In this section, based on a digital volcanic rock, parameters such as volume modulus, shear modulus, the stress–strain curve and porosity under different pressure conditions were calculated, respectively. We then tested and analyzed the feasibility of the finite element method by comparing the experimental parameters under the same temperature and pressure conditions. This part of the work lays the foundation for further study of the microscale influence of its mechanical properties. The common mechanical properties of rock are shown in Table 1 [49–51].

**Table 1.** Related parameters of the rock mechanics.

| Parameter | Numerical Value | Description |
|:---:|:---:|:---:|
| $a$ | 10 mm | Diameter of sample |
| $E_1$ | 30 GPa | Young's modulus |
| $\nu_1$ | 0.3 | Poisson's ratio |
| $\rho_1$ | 2.5 g/cm$^3$ | Density of sample |
| $c_1$ | 20 MPa | Cohesion |
| $\varphi_1$ | 35° | Angle of internal friction |

Volcanic rock is a kind of natural rock with strong heterogeneity, and its micro−fractures and pores are developed, which can actively reflect the deformation of the rock skeleton under the action of the pressure field. The lattice number of simulated volcanic rocks was $50 \times 50 \times 50$, and the voxel size was 2.64169 µm, while the real size is $1.32 \times 10^{-4}$ m. Its three−dimensional digital rock is shown in Figure 2a. Due to the influence of volcanic rock origin and time, the throat is relatively narrow and long. Macro−pores are relatively developed and complex in structure, while micro−pores are connected with macro−pores and are simple in structure. We used the same method of applying a series of displacements and calculating the stress–strain for the stress diagram of the whole geometry, as shown in Figure 2b. In order to view the stress distribution conveniently, the stress display range was reduced. It can be observed that the small structures close to the rock wall bear more stress and are prone to becoming the mechanical weak link.

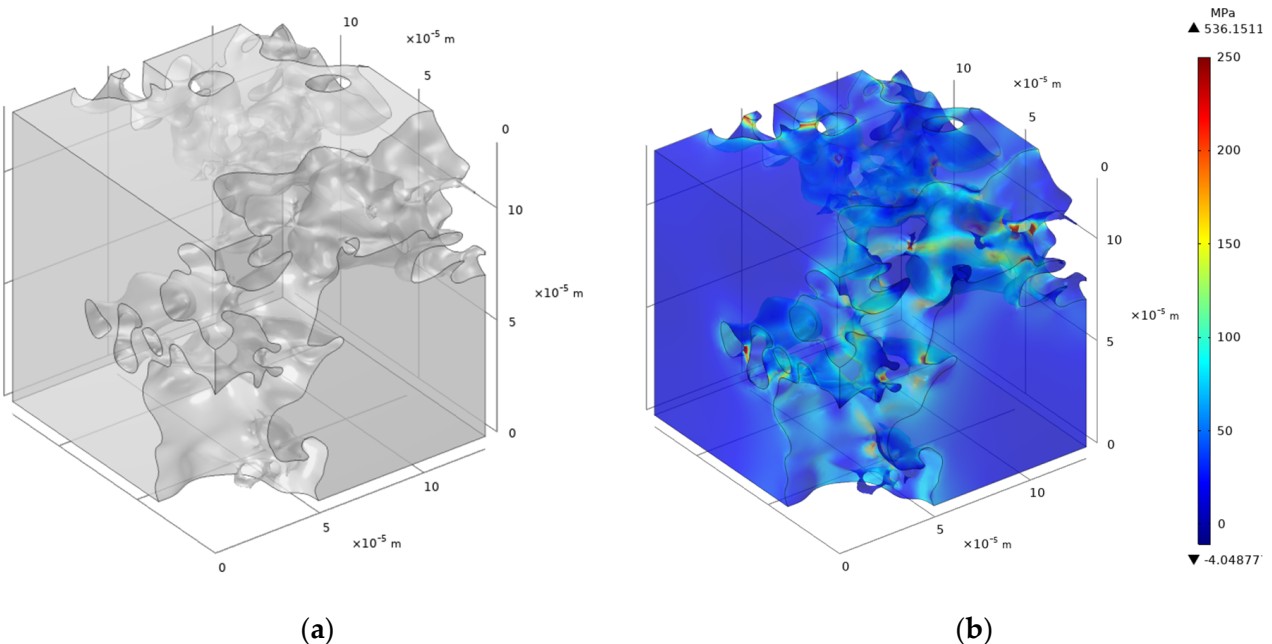

**(a)** **(b)**

**Figure 2.** 3D volcanic model image used in this study. (**a**) Represents the digital rock; (**b**) represents the resulting simulated strain diagram.

The calculated stress–strain curves are shown in Figure 3. Each curve is a conventional triaxial test, which is the stress change with a constant increase in the $z−$axis displacement

under the same *x* and *y* axis displacement. The slope of elastic regions could be obtained for verification. The elastic modulus obtained by calculation was 33.657 GPa. Through physical experiments, the volume modulus was 20.9 GPa and the Poisson's ratio was 0.228. The elastic modulus of 34.109 GPa was calculated by Equation (3), and the error was small. The simulation results were low, and the error may have been caused by the image coarsening and missing some of the small structures in the skeleton during image processing. In addition, there was a deviation between the experimental measurement position and the actual simulated position, and the scale was different, which also caused the error.

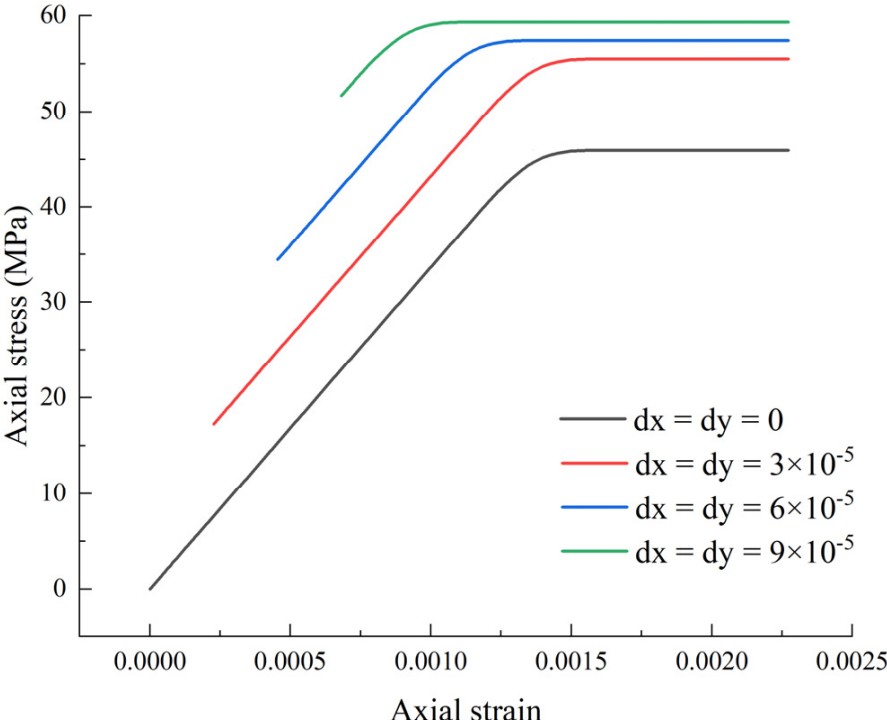

**Figure 3.** Stress–strain diagram of volcanic model, dx and dy is the *x* and *y* axial displacement.

Considering the complex structure of the actual rock, it is difficult to obtain effective quantitative rules. The sensitivity analysis was carried out by changing the rock structure model regularly. The laws of mechanical properties with particle size, fracture morphology, and clay content were obtained, which provide key basic data for further quantification of the fracturing modes.

### *3.2. Sensitivity Analysis*

### 3.2.1. Particle Size

As the main reservoir of oil and gas resources in the stratum, the properties of rock are complex. It is composed of particles of different minerals and different shapes through extrusion and cementation, so the mechanical properties of rock are largely affected by the mineral particles. The size of mineral particles is the primary factor affecting rock strength. According to the ball compact packing principle, the internal energy should be minimized to keep the ball in the most stable state, and the hexagonal closest packed method should be adopted, as shown in Figure 4. Spheres of different sizes were set up for simulation calculation to obtain the stress–strain diagram and elastic modulus, and the variation in strain with stress under the hexagonal tightest packed method was studied. The Young's modulus was solved, and the variation in elastic parameters under different particle sizes was studied by transverse comparison. The particle size parameters are shown in Table 2.

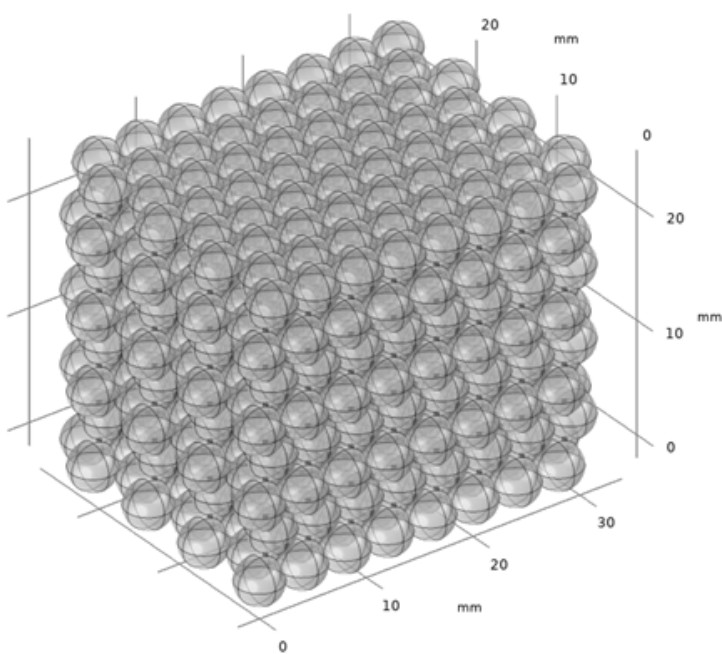

**Figure 4.** Model built using the hexagonal closest packed method.

**Table 2.** Particle size parameters related to changes in the mechanical parameters.

| Case No. | Case 1 | Case 2 | Case 3 | Case 4 | Case 5 | Case 6 | Case 7 |
|---|---|---|---|---|---|---|---|
| Particle size (mm) | 2 | 1 | 0.5 | 0.25 | 0.125 | 0.0625 | 0.01 |

Due to the strong similarity, Case 7 was selected for demonstration. Displacement was applied to the most marginal sphere in the $x$, $y$, and $z$ directions, the internal stress–strain was calculated, and the stress diagram was produced, as shown in Figure 5. As can be seen from the figure, the stress at the contact surface is large, which becomes the mechanical weak link prone to failure.

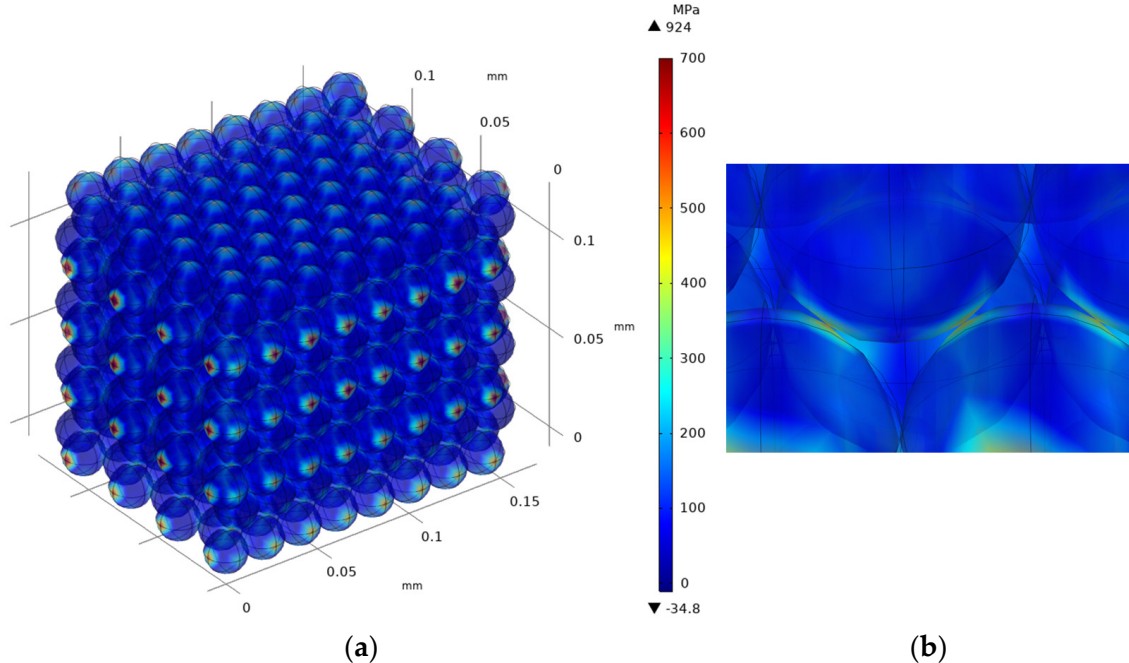

**Figure 5.** (**a**) Simulation strain diagram of Case 7, (**b**) the detailed diagram shows that the stress at the contact surface is large.

The stress–strain curve is shown in Figure 6. The linear area in the early stage of the curve is the elastic stage, the middle stage is the strengthening stage, and the late stage is the failure stage. Because the model is set to explore the influence of particle size on the rock mechanical properties, the elastic modulus fluctuation caused by a small contact area between particles was ignored. The model can still reflect the change trend of peak stress and geometric structure, which lays the foundation for further study on the influence of particle size.

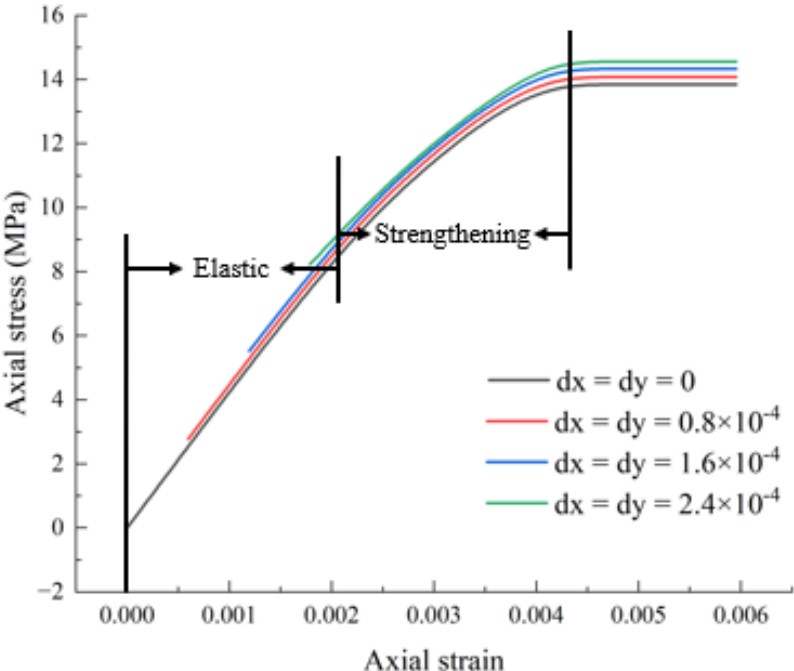

**Figure 6.** Simulation stress–strain curve of Case 7, where the elastic stage and strengthening stage can be seen.

The strain data were further processed to study the effect of stress on their porosity. The variation in porosity with stress is shown in Figure 7. It can be seen from the figure that the forced displacement increases the stress, thus changing the strain of the whole geometric structure. With the increase in the stress, the porosity of the particle model gradually decreases. In the early stage, when the displacement is applied in the $x$, $y$, and $z$ directions, the porosity decreases linearly. The downward trend of porosity gradually slows down in the middle and late stages due to the gradual cessation of the pressure in the $x$ and $y$ axis while the $z$ axis pressure continued. This shows that stress and porosity have a good linear relationship.

In order to quantitatively characterize the 3D pore structure under triaxial pressure, a topologically equivalent 3D pore network model was established based on the simulation results of different particle sizes. In order to characterize the pore structure and study the influence of stress on the pore−throat parameters, the total pore volume, average pore equivalent radius, average throat equivalent radius, average pore–throat ratio and average coordination number were measured and recorded. The variation curve of the pore throat parameters with stress is shown in Figure 8.

When the volumetric strain was between 0.0108 and 0.0157, the pore coordination number, throat equivalent radius, throat length, and other pore and throat parameters changed abruptly in response to the volumetric strain. It can be concluded that there may be a critical value related to the stress and pore structure that controls the sudden change in strain. The same conclusion was recently reported by Yang Ju et al. [52].

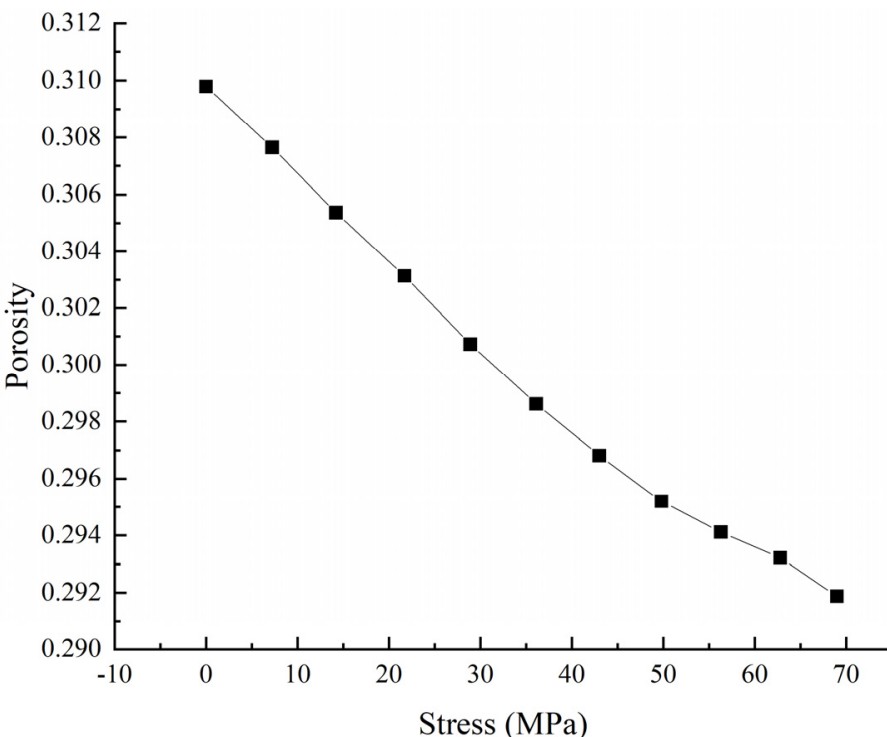

**Figure 7.** The porosity decreases as the stress increases in the same porous medium.

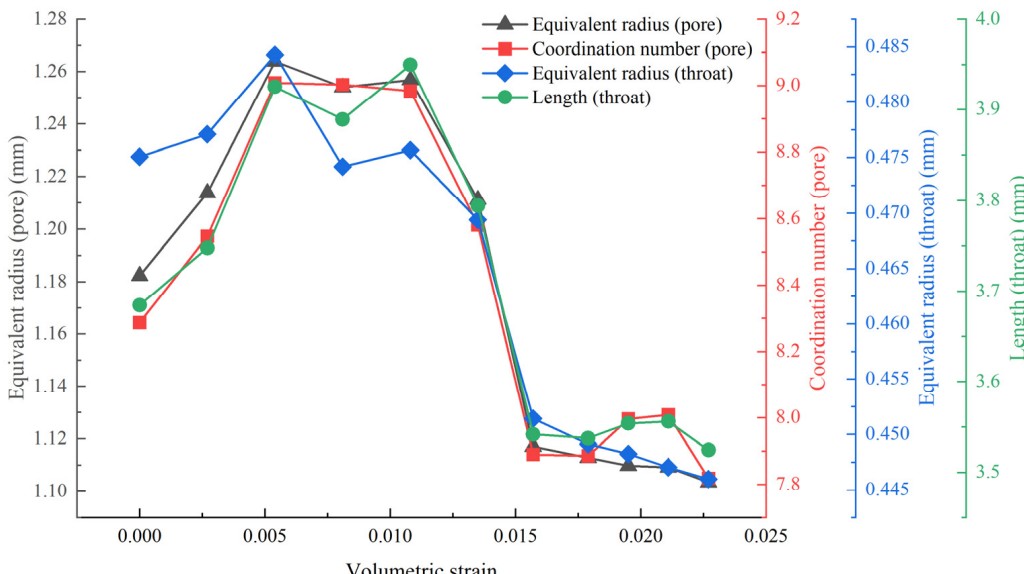

**Figure 8.** The pore and throat parameters will change abruptly in response to volumetric strain.

The influence of particle size on rock strength was studied by repeating the above simulation with the change in particle size. The curve of the stress with particle size is shown in Figure 9. As can be seen from the figure, when the particle size was small, the rock strength was higher and more sensitive; when the particle size was large, the rock strength was low and not easily affected by the particle size. This is because the particle size was small, the whole geometric deformable range was small, and the pore structure strain was not obvious.

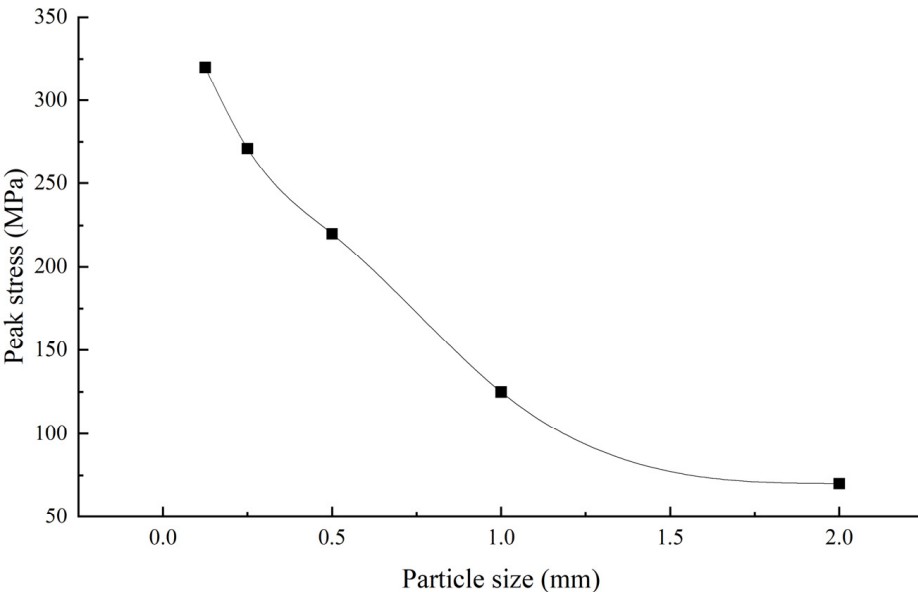

**Figure 9.** When the particle size is large, the peak stress is small and sensitive.

### 3.2.2. Fracture Morphology

Another important factor affecting the mechanical properties of rock is the fracture, which can change its shape to cope with the strain during the process of rock compression. In order to explore the influence of fracture morphology on the rock mechanical properties, fractures with different angles and apertures were set for numerical simulation. Specific parameters are shown in Table 3.

**Table 3.** Combinations of the different fracture parameters used to reflect the fracture morphology.

| Case | Geometric Parameters of Prefabricated Fractures | |
| --- | --- | --- |
| | Quantification | Variable |
| Fracture angle | $b = 1$ | $\alpha = 0°, 15°, 30°, 45°, 60°, 75°, 90°$ |
| Fracture aperture | $\alpha = 60°$ | $b = 0.01$ mm, $0.1$ mm, $0.5$ mm, $1$ mm, $2$ mm |

The influence of the fracture angle on the peak stress and elastic modulus is shown in Figures 10 and 11. As can be seen from the figure, when the angle between the fracture plane and the rock cross section (which is the plane perpendicular to the principal stress) was less than 45° or more than 75°, the peak stress changed little. This is because the angle is small, and the main stress surface is the plane where the fracture is located, so the stress can be distributed evenly and improve the strength of the sample. When the angle is larger, the fracture and axial stress tend to be more parallel, and the fracture force is smaller. When the fracture angle is between 45° and 75°, the fracture has a great influence on the peak stress. This is due to the inclination of the fracture, the uneven distribution of forces on both sides of the fracture, and the shear of the sample is easier to breed and evolve. The influence trend of fracture angle on the elastic modulus of the model is the same as that of the peak stress, but it was smoother and closer to a linear trend. This suggests that during fracturing, the fracture angle between natural fractures and the fracturing direction should be less than 45° as much as possible. On the one hand, the smaller fracture angle makes it easier for the fracturing fractures to connect the natural fracture; on the other hand, as stress reduces the peak pressure and elastic modulus of the rocks around the natural fractures, deterioration of the rock mechanical properties is more likely to cause rock failure.

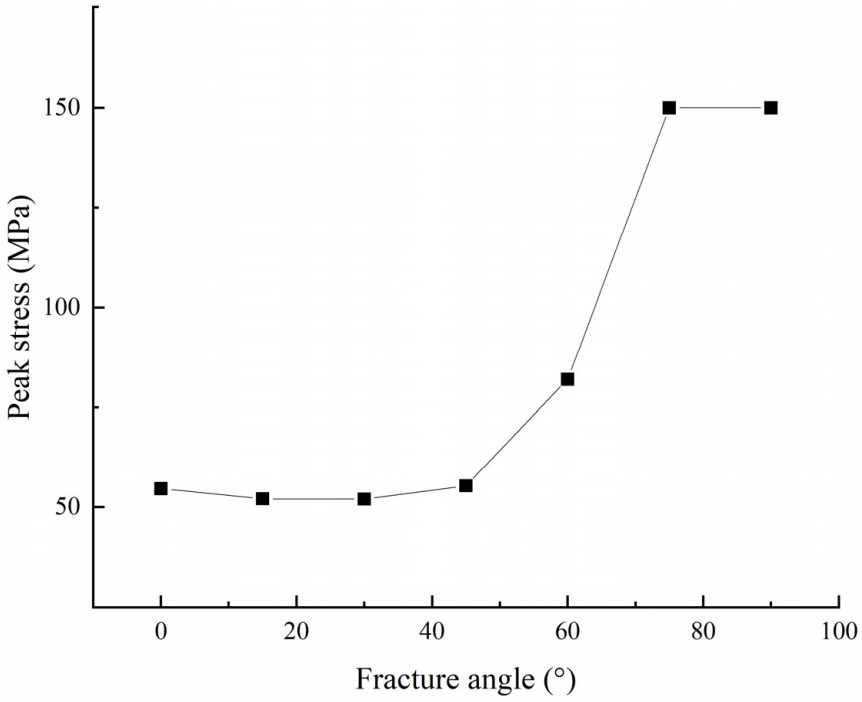

**Figure 10.** When the fracture angle is between 45° and 75°, the peak stress changes greatly.

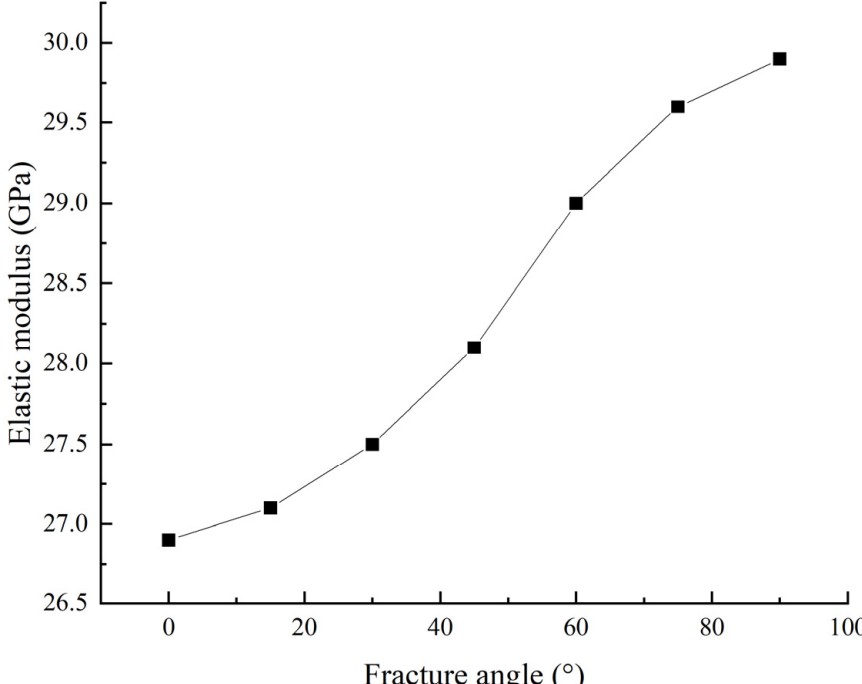

**Figure 11.** The elastic modulus increases with the fracture angle, which is smooth and linear.

In order to eliminate data errors caused by too sharp cuboid boundaries, the fracture was set as ellipsoid. The current fracture angle was 60°. The stress distribution of this model is shown in Figure 12. The lower part of the fracture was the area with large stress, and the part above the plane where the fracture is located was the main source of stress.

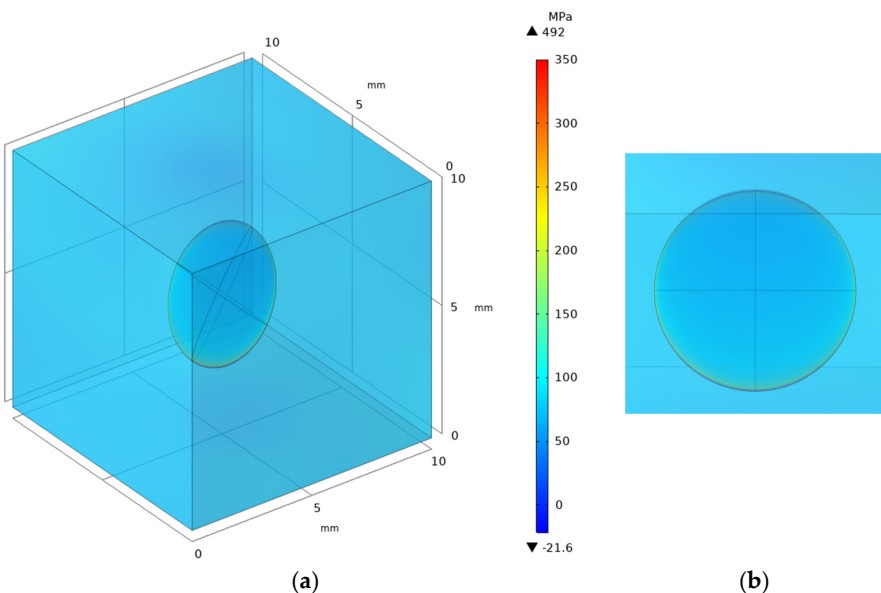

**Figure 12.** (**a**) Simulation strain diagram of ellipsoidal fracture, (**b**) the detailed diagram shows the stress at the fracture.

The influence of the fracture aperture on the peak stress and elastic modulus is shown in Figures 13 and 14. When there is no fracture or the fracture aperture is zero, the peak stress of the model is higher. The elastic modulus is 30 GPa, consistent with the model input shown in Table 1. Once the fractures are added, the peak pressure of the sample decreases abruptly, and gradually increases with the increase in the fracture aperture. This is because the fractures are too small, and the edge structure is sharp and easy to damage. With the increase in the fracture aperture, the fracture edge gradually becomes smooth and the sample strength is greater. When the fracture aperture is 2 mm, the peak stress decreases because the fracture aperture will eventually reduce the rock strength. The elastic modulus decreases abruptly with the addition of fractures and becomes stable with the increase in the fracture aperture. This suggests that the sharper the fracture edges, the more prone to fracture propagation. When the fracture aperture is large enough, the reduction in the elastic modulus will greatly reduce the rock strength. Of course, the problem of fracture roughness was not considered here. When the fracture aperture is small but the roughness is large, the sudden change in mechanical properties may not occur.

3.2.3. Clay Content

Although the clay still bears some stress, it is fundamentally different from the rock skeleton, and the rock mechanical properties deteriorate with the increase in the clay content. John W. Minear proposed that the clay distribution forms were divided into three categories according to their elastic characteristics and contributions to the mechanical parameters: dispersed clay dispersed in pores, not under pressure from the rock skeleton, can be regarded as pore fluid; laminated clay subjected to rock pressure; structural clay is distributed like a particle in the rock [53]. In order to comprehensively consider the effect of clay content on the rock mechanical properties, the clay without a mechanical effect was removed, and the laminated clay and structural clay with clay contents of 0.25, 0.125 and 0.375 were simulated. The parameter setting of clay is shown in Table 4 [54].

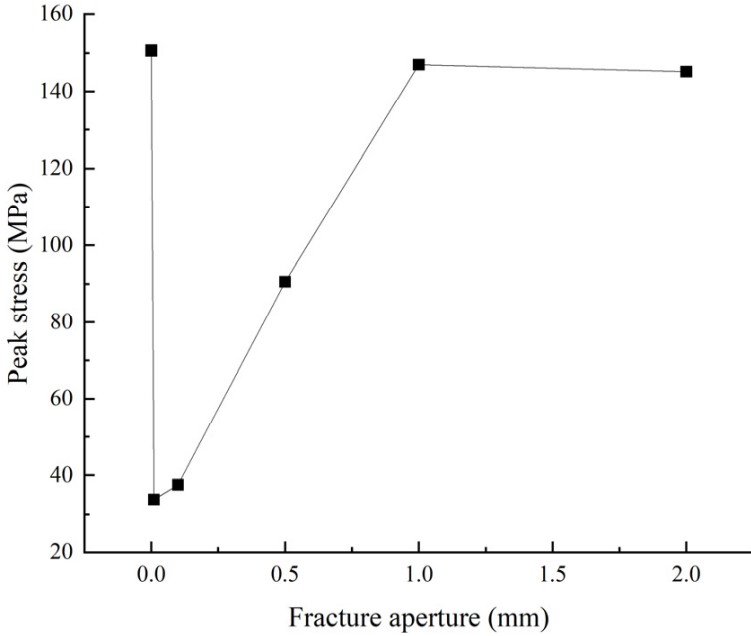

**Figure 13.** The peak stress is greatly affected when the fracture aperture is extremely small, and the peak stress decreases gradually when the fracture aperture is greater than 2 mm.

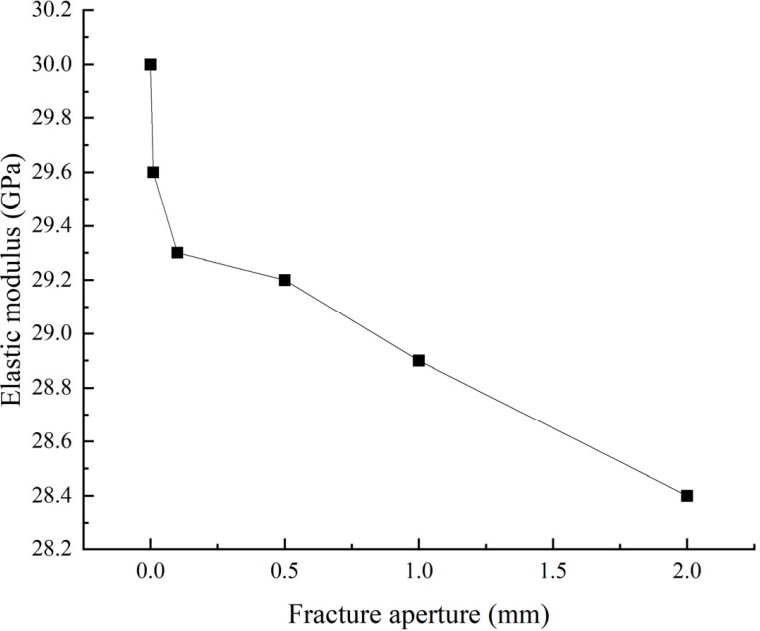

**Figure 14.** The elastic modulus decreases abruptly with the addition of fractures and becomes stable when the fracture aperture is large enough.

**Table 4.** Related parameters of the rock mechanics.

| Parameter | Numerical Value | Description |
|---|---|---|
| $r$ | 10 mm | Diameter of particle |
| $E_2$ | 23 GPa | Young's modulus of clay |
| $v_2$ | 0.34 | Poisson's ratio of clay |
| $\rho_2$ | 2.55 g/cm$^3$ | Density of clay |
| $c_2$ | 10 MPa | Cohesion of clay |
| $\varphi_2$ | 30° | Angle of internal friction of clay |

Considering that laminated clay can be divided into the xy plane, xz plane, and yz plane according to the plane position of the clay layer, in order to simplify the calculation steps, only the laminated clay in the xy and xz planes was simulated, and the yz plane with a similar structure to the xz plane was not simulated. The clay models are shown in Figure 15.

The relationship between the clay content and the peak stress is shown in Figure 16. When the structural clay content is small, the peak stress of rock is still at a high level, but the sensitivity is high. This is because the addition of clay greatly affects the rock strength. When the clay content is higher than 25%, the peak stress varies slightly. This is because the rock skeleton plays a major role in bearing the pressure, and the distributed clay stress is small. This means that when the structural clay content is greater than 25%, the strength of the rock has been reduced to a relatively low level, making fracturing easier.

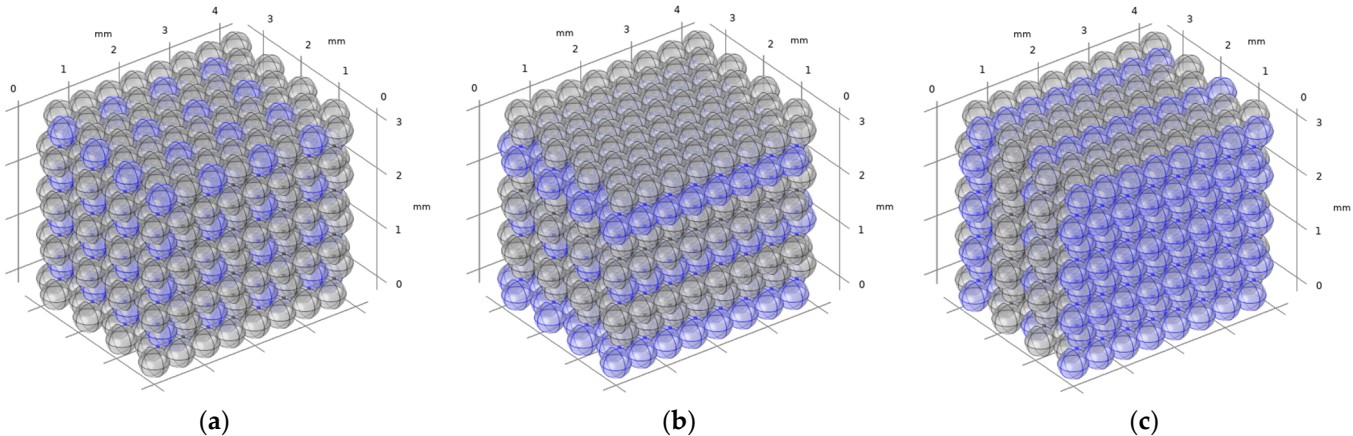

(**a**) (**b**) (**c**)

**Figure 15.** Model to examine the effect of clay content, the blue areas are clay. (**a**) Represents the structural clay; (**b**) represents the laminated clay in the xy plane; (**c**) represents the laminated clay in the xz plane.

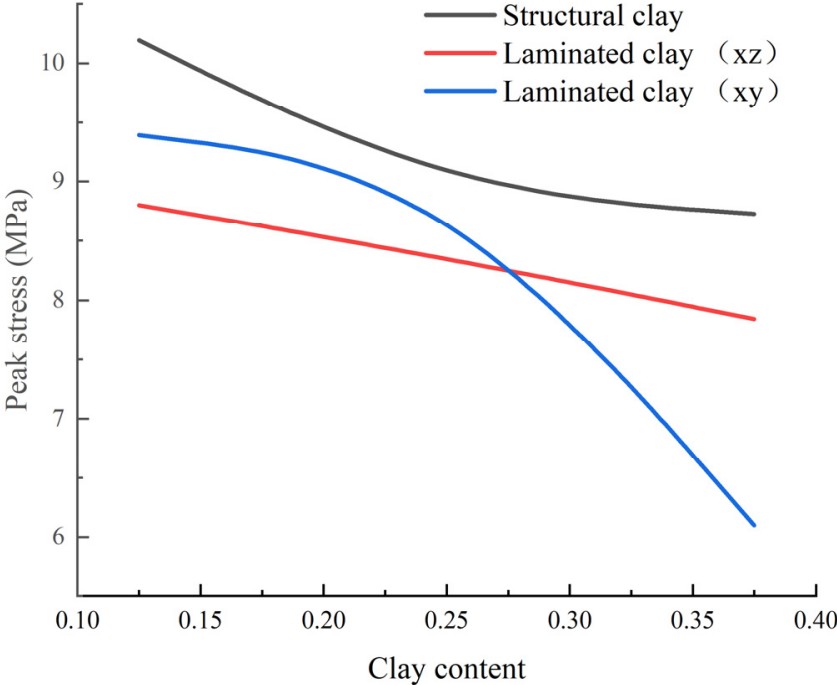

**Figure 16.** Peak stress vs. content of different clays.

The effect of the xy plane laminated clay content on the peak stress is opposite to that of the structural clay content. The peak stress changed little when the clay content was low, but decreased rapidly when the clay content was high. This is because the xy plane is perpendicular to the stress direction. The laminated clay has a large internal homogeneity and can better complete the work of bearing pressure. However, the increase in clay content will eventually greatly affect the rock strength.

The effect of the xz plane laminated clay content on the peak stress was linear. This is because the xz plane is parallel to the stress direction, and the rock skeleton under the main stress can be connected. The "force chain" is relatively complete, and the peak stress decreases with the reduction in the "force chain".

The laminated clay is distributed in sheets, and its peak stress is lower than that of structural clay at all clay contents. When the clay content is small, the force surface of the xy plane laminated clay is large and uniform, and its peak stress is less affected. However, the "force chain" in the xz plane laminated clay continues to lose and its peak stress gradually decreases. However, when the clay content is greater than 27%, the large amount of clay causes the peak stress of the whole model to decrease substantially. For the xz plane laminated clay with stable "force chain", its peak stress still showed a linear decreasing trend.

Studies of the clay content showed the following: the strength of structural clay rocks is higher than that of laminated clay rocks with the same content of clay; when the clay content is less than 27%, the strength of the xz plane laminated clay rocks is lower than that in the xy plane; and when the clay content is greater than 27%, the strength of the xy plane laminated clay rocks is higher. This suggests that it is easier to fracture in a direction parallel to the laminated clay when the clay content is below 27%. Conversely, the direction perpendicular to the laminated clay rocks is better.

## 4. Conclusions

In this work, based on the digital rock, true triaxial test numerical simulation experiments were conducted on representative rocks and the natural pore structure. This work highlights the effects of rock particles, fractures, and clay on the mechanical parameters. The main findings from the study are:

1.  The simulation method can simulate the true triaxial experiment well and is in good agreement with the experimental results;
2.  The stress at the contact surface is large, which becomes the mechanical weak link prone to failure. The porosity decreases with the increase in stress and has a good linear relationship. However, when the particle model volumetric strain is between 0.0108 and 0.0157, the pore and throat parameters will change abruptly. When the particle size is small, the rock strength is higher and more sensitive;
3.  When the fracture angle is between 45° and 75°, the fracture has a great influence on the peak stress. The angle between the natural fracture and the fracturing direction should be less than 45° as much as possible. Sharper fracture edges are more prone to fracture propagation;
4.  Clay affects the rock strength by influencing the force chains formed by the rock skeleton. The peak stress is gradually decreased with the increase in the content of clay, but the influence of different types of clay is different. Fracturing is easier when the structural clay content is higher than 25%. It is easier to fracture in a direction parallel to the laminated clay when the clay content is below 27%. Conversely, the direction perpendicular to the laminated clay rocks is better.

The results of this study provide key fundamental data for further quantifying fracturing patterns and guiding hydraulic fracturing to promote oil and gas well stimulation. However, the limitation of this work is that it is highly targeted and does not involve microscopic factors such as heterogeneity, complex rock structure, and fluid in porous media.

**Author Contributions:** Conceptualization, W.M. and Y.Y.; Methodology, W.M.; Validation, W.M. and J.Y. (Jiangshan Yang); Formal analysis, W.M. and W.Y.; Writing—original draft preparation, W.M.; Writing—review and editing, Y.Y., W.Y. and C.L.; Supervision, W.S.; Project administration, W.S., H.S., L.Z., K.Z. and J.Y. (Jun Yao). All authors have read and agreed to the published version of the manuscript.

**Funding:** This research was funded by the National Natural Science Foundation of China (No. 52034010), the Shandong Provincial Natural Science Foundation (No. ZR2019JQ21), and the Program for Changjiang Scholars and Innovative Research Team in University (IRT_16R69).

**Data Availability Statement:** Publicly available datasets were analyzed in this study. These data can be found here: https://pan.baidu.com/s/1xUEp9_A2-mRbZrOLWOF2Ow?pwd=0000, accessed on 16 January 2023.

**Conflicts of Interest:** The authors declare no conflict of interest. The funders had no role in the design of the study; in the collection, analyses, or interpretation of data; in the writing of the manuscript; or in the decision to publish the results.

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
