# Peer review of "Digital Rock Mechanical Properties by Simulation of True Triaxial Test: Impact of Microscale Factors"

_geotechnics, doi:10.3390/geotechnics3010002_

Round 1

Reviewer 1 Report

Dear Editor and Authors.

It has been my pleasure to review the paper “Digital rock mechanical properties by simulation of true triaxial test: impact of microscale factors”. In this manuscript, the authors present a study provides fundamental data, based on numerical simulation, the key to further quantifying fracturing patterns, indicating that the effects of rock particles and clay fractures on mechanical parameters. For this purpose, accurate triaxial tests had been simulated in representative rocks and the natural pore structure. The force state closest to the actual situation and exploring the change rule had been simulated at the microscale level.

The topic of this paper could be appropriate to the scope of Geotechnics. However, some mistakes should be corrected before. These suggestions and corrections are listed below:

1. Line 60. It is suggested that the not-so-formal linking word "so" be replaced by something more formal such as “thus”, “hence”,…

2. Line 174. Remove the mathematical formula from equation 3, the reference to equation 3 is sufficient.

3. Line 200. The use of bold type for the reference in Table 2 is not correct.

4. Line 204 and others such as Figure 5 and Figure 6. Case7 appears whereas in Table 2 it appears as Case 7. Please correct this so that there is only one single format.

5. Lines 274 to 277. Rewrote the sentence. It is a bit confusing/unclear.

6. Line 317. Unnecessary capital letters in “Laminated clays” and “Structural clay” after the semicolon.

7. Line 359. Extra space at the beginning of the sentence.

Some figure and table labels are very poor. Examples are: Table 2, Figure 7, Figure 8, Figure 9, Figure 10, Figure 11, Figure 12, Figure 14, and Figure 15. It is highly recommended to use more clarifying, descriptive and appropriate labels for a scientific publication.

Finally, Figure 12 should be considered for redundancy. It does not seem to be necessary considering the information provided in Figure 13.

For my part, I consider the manuscript may be considered for publication after a review of the original manuscript. I insist on a pleasure review of the manuscript submitted by the authors. I look forward to hearing from you.

Sincerely,

Reviewer.

Author Response

Response to Reviewer 1 Comments

We gratefully thank the editor and all reviewers for their useful remarks and suggestions to improve the paper quality. We have carefully made modifications according to the reviewers’ suggestions and addressed all the reviewers’ concerns. The detailed response is given below and marked in blue.

Reviewer #1: It has been my pleasure to review the paper “Digital rock mechanical properties by simulation of true triaxial test: impact of microscale factors”. In this manuscript, the authors present a study provides fundamental data, based on numerical simulation, the key to further quantifying fracturing patterns, indicating that the effects of rock particles and clay fractures on mechanical parameters. For this purpose, accurate triaxial tests had been simulated in representative rocks and the natural pore structure. The force state closest to the actual situation and exploring the change rule had been simulated at the microscale level.

The topic of this paper could be appropriate to the scope of Geotechnics. However, some mistakes should be corrected before. These suggestions and corrections are listed below:

  1. Line 60. It is suggested that the not-so-formal linking word "so" be replaced by something more formal such as “thus”, “hence”,…

Response:

Thank you for your careful check, and we have proofread the manuscript and modified all these not-so-formal linking word errors.

  1. Line 174. Remove the mathematical formula from equation 3, the reference to equation 3 is sufficient.

Response:

We are sorry for this formatting error due to cross-reference error, and we have proofread the manuscript and modified all these cross-reference errors.

  1. Line 200. The use of bold type for the reference in Table 2 is not correct.

Response:

We are sorry for this format problem during the full text check, and we have modified it.

  1. Line 204 and others such as Figure 5 and Figure 6. Case7 appears whereas in Table 2 it appears as Case 7. Please correct this so that there is only one single format.

Response:

We appreciate this valuable suggestion, and we have proofread the manuscript and modified all the problem of inconsistent format errors.

  1. Lines 274 to 277. Rewrote the sentence. It is a bit confusing/unclear.

Response:

Thank you very much for the good suggestion. In fact, there is an analysis of why the fracture angle should be less than 45° as much as possible. On the one hand, the smaller fracture angle makes it easier for the fracturing fractures to connect the natural fracture; on the other hand, stress reduces the peak pressure and elastic modulus of the rocks around the natural fractures, deterioration of rock mechanical properties is more likely to cause rock failure. We have defined the description of the fractures and the whole sentence expression in the revised manuscript.

  1. Line 317. Unnecessary capital letters in “Laminated clays” and “Structural clay” after the semicolon.

Response:

Thank you for your careful check on punctuation format, and we have proofread the manuscript and modified such errors.

  1. Line 359. Extra space at the beginning of the sentence.

Response:

Thank you for your careful check, and we have proofread the manuscript and modified sentence format errors.

Some figure and table labels are very poor. Examples are: Table 2, Figure 7, Figure 8, Figure 9, Figure 10, Figure 11, Figure 12, Figure 14, and Figure 15. It is highly recommended to use more clarifying, descriptive and appropriate labels for a scientific publication.

Response:

This suggestion is appreciated. We have enriched labels in the revised manuscript. A scientific article should indeed have more clarifying, descriptive and appropriate labels.

Finally, Figure 12 should be considered for redundancy. It does not seem to be necessary considering the information provided in Figure 13.

Response:

Thank you for pointing out the redundancy in Figure 12. In fact, we want to be more complete and ignore the problem of repetition. We have proofread the manuscript and modified redundancy errors.

We appreciate the comments of the reviewers earnestly and hope that the responses will meet with approval.

Thanks for your valuable comments and suggestions. We have carefully revised the manuscript and the reviewers’ comments are responded to one by one. If there are some answers which are not clear, we would like to explain again. We hope that the revised version can meet with approval. Thank you for your time and attention!

Reviewer 2 Report

The work submitted to the MDPI Geotechnics journal entitled as “Digital Rock Mechanical Properties by Simulation of True Tri-axial Test: Impact of Microscale Factors” is reviewed. 

This study presents a new technique based on the digital rock, true triaxial test numerical simulation experiments conducted on representative rocks and natural pore structure. The rock mechanical properties are studied on the microscale to simulate the force state closer to the real situation and explore the change rule. The reviewer believes this research paper could be an interesting to petroleum and natural gas and civil engineering research community and those who are interested in rock mechanics.

In general, paper is well structured, and the data is well analysed and requires minor revision to be evaluated. I am suggesting the manuscript to be accepted for publication from the MDPI Geotechnics however, if the authors are willing to perform major improvements / corrections on the submitted work. 

Here are the major improvements / corrections I suggest authors to review:

·       L 25, L 132 – Do not use personal pronouns in scientific writing.

·       L 72 - Is it only clay content deteriorates strength of rock? What about silt? Please add a few references in respect to effect of silt bands in rock mass.

·       Indicate the name of the software packages used for the simulations.

·       How did authors introduce the pore pattern within the Rock mass shown in the Figure 2. 

·       L176 Authors are proposing the reason of observed difference in measured parameters are due to image coarsening However they haven’t introduced this concept image processing earlier by describing how each process was done?

·       L193 – Authors should state what is the difference of adapted me method in respect to Discrete Element Modeling.

·       L 230 – Discuses the observed porosity vs stress behavior in respect to a experimental result of rock/ soil specimen.

·       L 244 – Refer to guidelines in respect to in text referencing “Dr. Ju”

·       L246 – What would happen to Peak stress if well graded (same percentage of each particle size” sample is tested?

·       Figures 10 and 11 rewrite the caption of the figures to be more descriptive.

·       L 349 Discuss why the peak stress clay laminated in xy direction is experiencing reduction after 27% of content and xz is not?

·       General Comments – Revise the keywords according to journal guidelines.

·       General comment – there is complexity on the purpose of this particular research. It should be uniquely stated what the aim is then the importance should be appreciated by those who are interested in this paper.

·       General Comments – Conclusions sections should be re-arranged as Conclusion and Recommendations. In this section limitations and recommendations of this study should be listed.

·       General Comments – There are some of grammatical mistakes and drawbacks in the manuscript, Please improve the English and try to present a concise expression.

·       General comment – References section should be reviewed as few references are not according to the journal guidelines. 

Author Response

Please see authors response in attachment

Round 2

Reviewer 2 Report

The work submitted to the MDPI Geotechnics entitled "Digital Rock Mechanical Properties by Simulation of True Triaxial Test: Impact of Microscale Factors" is re-reviewed.

In general, the authors have successfully answered and reflected most of the concerns arise in the previous round of review. I am suggesting the manuscript be accepted as is for publication.